# Glycoprotein Production by Bursal Secretory Dendritic Cells in Normal, Vaccinated, and Infectious Bursal Disease Virus (IBDV)-Infected Chickens

**DOI:** 10.3390/v14081689

**Published:** 2022-07-30

**Authors:** Balázs Felföldi, Zsófia Benyeda, Tamás Kovács, Nándor Nagy, Attila Magyar, Imre Oláh

**Affiliations:** 1Ceva-Phylaxia Ltd., 1107 Budapest, Hungary; balazs.felfoldi@ceva.com; 2Biovo Animal Health Ltd., 7700 Mohács, Hungary; zsbenyeda@prophyl.hu; 3Department of Anatomy, Histology and Embryology Semmelweis University, 1094 Budapest, Hungary; kovacs.tamas@med.semmelweis-univ.hu (T.K.); nagy.nandor@med.semmelweis-univ.hu (N.N.); magyar.attila@med.semmelweis-univ.hu (A.M.)

**Keywords:** bursal secretory dendritic cell (BSDC), glycoprotein (gp), IBDV infection, vaccination effect on gp production

## Abstract

The aim of this study is to follow the gp production in IBDV-vaccinated and challenged birds. The progress of IBDV infection was monitored using anti-VP2 immunocytochemistry, light and transmission electron microscopy. In the medulla of the bursal follicle, the Movat pentachrome staining discovered an extracellular glycoprotein (gp) produced by bursal secretory dendritic cells (BSDCs). The secretory granules of BSDCs either discharge resulting in extracellular gp or fuse together forming intracellular corpuscles. The double fate of granules suggests a dual function of BSDCs: (a.) For the discharged granules, gp contributes to the medullary microenvironment (ME). (b.) The intracellular corpuscles may be the sign of BSDC transformation to a macrophage-like cell (Mal). Infectious bursal disease virus (IBDV) infection accelerates the BSDC transformation to Mal. The decreased number of BSDCs is feedback for the precursor cells of BSDCs lodging in the cortico-medullary epithelial arches (CMEA), where they proliferate. Opening the CMEA, the precursor cells enter the medulla, and differentiate to immature BSDCs. The virus uptake in the corpuscles prevents the granular discharge resulting in the absence of gp and alteration in ME. In vaccine-take birds, the mitotic rate of BSDC precursor cells cannot restore the precursor pool; therefore, in the case of IBDV challenge, the number of newly formed BSDCs is too low for outbreak of clinical disease. The BSDCs, as a primary target of IBDV, may contribute to the long-lasting immunosuppressive status of IBDV-infected chickens.

## 1. Introduction

Chickens are highly susceptible to infectious bursal disease virus (IBDV) during their whole life; however, clinical disease occurs most frequently between 3 and 17 weeks of age, which is similar to the steady state condition of bursa of Fabricius (BF). It has been reported that in newly hatched chickens the virus can replicate without clinical symptoms [1]. The age-related susceptibility may be dependent on the lymphocyte population in the medulla of bursal follicles [1,2,3,4,5,6]. It is suggested that a large number of highly susceptible cells, lymphocytes, are crucial for clinical IBDV infection [1,4,7]. Okoye et al. [8] also declared that “healthy bursa is essential for the development of clinical disease (IBD)”. This statement is based on experiments, in which embryonal bursectomized and cyclophosphamide (Cy)-treated chickens do not develop clinical IBD; both experiments result in B cell depletion. The IBDV is bursa specific; therefore, Hitchner [9] suggested that in adult chicken clinical IBD does not develop, because of normal involution of BF. However, Rautenschlein et al. [10] did not find age-related differences in outbreaks of clinical disease between 12- and 28-day-old birds, but they observed tissue-specific differences in cytokine production (IFNγ and IL-1β). In ovo vaccinated birds (18 days of incubation) recovered from bursal lesion faster than post-hatched vaccinated chickens [11].

The maternally derived antibodies (MDAs) are insufficient to protect against disease, but not for bursal lesions [12]. The MDAs interfere with vaccination [13,14,15], namely, if a bird is receipted for live virus vaccination, the individual MDA level may prevent the replication of vaccine virus, while not sufficient to prevent field infection, as the breakthrough titers of different IBDV strains show considerable variations. There is a critical period, the length of which may be different from chick to chick, when a chick can be infected but is still refractory to immunization [12].

In the medulla of bursal follicle—where the first IBDV-infected cells appear—the BSDC is the only secretory cell type, which produces different-sized, electron-dense granules. The granules either discharge [16] or fuse together forming large, dense, intracytoplasmic bodies [17]. The formation of intracytoplasmic bodies, corpuscles, is possibly the first sign of BSDC transformation to macrophage-like cells (Mal) [17]. In normal birds, the Mal is the terminal maturation of BSDCs, which migrate out from the medulla to the follicle-associated epithelium (FAE) and occasionally to the cortex. The discharged granular substance (by exocytosis) attaches in a high concentration to the cell membrane of BSDCs and appears as a membrane-bound substance [16]. The membrane-bound substance can be gradually detached from the cell membrane and a fine, flocculated substance appears in the extracellular space of the medulla. Movat pentachrome staining shows that the solubilized substance is a glycoprotein (gp), which fills up the intercellular space of follicular medulla [17,18]. The gp may provide a unique microenvironment (ME) for medullary, but not cortical, B cells. This secreted, flocculated, and solubilized substance is bursa-specific, and can bind to medullary lymphocytes [19].

It is generally accepted that the major target cell of IBDV is the bursal IgM-positive, medullary B cell [1,20,21,22], but IBDV can replicate in lymphocytes, macrophages, and granulocytes [4,6,23,24,25], but not in “immature” lymphoblasts [26]. The “immature” lymphoblast-like cells lodge in the cortico-medullary epithelial arches (CMEA) [27] and do not express chB6-positive membrane antigens and surface IgMs. In the medulla, the Mal, which is actually a transformed BSDC, pack up the virus particles in the former fused BSDC granules. Packaging the virus particles inhibits the granular discharge of BSDCs resulting in extracellular shortage or absence of gp in the medulla. The altered medullary ME subsequently results in lymphocyte death [17].

In non-vaccinated birds, the IBDV infection abolished the Movat-positive, extracellular gp in the medulla, but gp emerged in the Mal [17,18]. The aim of this paper is to provide evidence for the effect of IBDV infection on the gp producing BSDCs and the alteration of Movat-positive, bursa-specific gp in infected, vaccinated, and vaccinated plus challenged chickens.

## 2. Materials and Methods

### 2.1. Animals

White Leghorn layer type chickens of SPF status were used in the trials. The chickens were hatched and reared at Ceva Phylaxia Animal housed for 4 weeks under SPF conditions. The chickens were randomly placed into four groups: non-vaccinated non-challenged (NV-NC), non-vaccinated challenged (NV-C), vaccinated non-challenged (V-NC), and vaccinated and challenged (V-C) groups (Table 1).

After live IBDV vaccination at 4 weeks of age, all groups were placed into isolator units and kept there until termination. Conditions were set according to the need of chickens: 26 °C ambient temperature, 150 mPa overpressure, 30–40 m^3^/h air exchange, 12–12 h light/dark cycle; food (sterilized commercial broiler grower feed) and water (tap water) were available ad libitum; environmental enrichment was provided by ladders and sand containers. During the trial, all actions performed conform to the regulation of the EU decree No. 40 of 2013. The chickens were euthanized by overdosed pentobarbital injection at 4 days post-infection (35 days of age). In the NV-C group, additional sampling was included at 36 and 48 h post-infection to monitor the progression of IBDV infection.

After challenge infection, chickens were frequently monitored for IBDV clinical symptoms and a humane endpoint was provided to moribund specimens by overdosed pentobarbital injection.

### 2.2. IBDV Strains

For vaccination, the 228E-attenuated classical IBDV strain was used at a dosage of 3.0 lgEID50/chicken, diluted in 0.2 mL sterile PBS, applied individually per-os. Vaccination was performed at 28 days of age in V-NC and V-C groups. In NV-C and V-C groups, the challenge infection was performed at 31 days of age (3 days post-vaccination), by the time the vaccine virus replication spread through it was spreading through the entire bursa. Challenge strains used in the trial were the very virulent IBDV isolate D407/2/04/TR [28] and variant Delaware-E ([29]; [GenBank: AF133904]) strains.

According to current classification of IBDV [30] the 228E strain [GenBank: AF457104] belongs to genogroup 1 (classical IBDV), Delaware-E strain to genogroup 2 (antigenic variant IBDV), and D407/2/04/TR to genogroup 3 (very virulent IBDV), based on genetic analysis of the hypervariable region of the VP2 gene. The pathogenicity of challenge strains was characterized by previous trials at Ceva-Phylaxia Ltd. Budapest, Hungary and the D407/2/04/TR isolate was found to cause clinical symptoms, high mortality, and serious bursal lesions in susceptible chickens [28], consistently with vvIBDV, while the Delaware-E strain (reference strain of the antigenic variant group) caused milder bursal lesions and no clinical symptoms (unpublished data).

Each strain was used by dosage of 4.0 lgEID50/chicken, applied individually per-os, diluted in 0.2 mL sterile PBS. Virus strains (vaccine and field) were propagated and titrated in embryonated SPF chicken eggs; the end point titer was determined using the Spearmann–Kaerber method.

### 2.3. Antibodies

The anti-IBDV monoclonal antibody (mAb) 5A10 was received from Ceva-Phylaxia, Hungary. The antibody was produced against the VP2 protein of IBDV. Cleaved (active) Caspase3 mAb (Cell Signaling No9660) was used for identification of apoptotic cells.

### 2.4. Immunocytochemistry for Demonstration of IBDV-Infected Cells

From each chicken, the tissue samples were embedded in their own liver and frozen in liquid nitrogen. The blocks were stored in a capped tube at −80 °C until cryostat sectioning. The 10 µm cryostat sections were fixed in cold acetone for 10 min and rehydrated in PBS. The sections were incubated with primary antibody for 45 min at room temperature. After washing, the isotype-specific biotinylated secondary antibody was used. The endogenous peroxidase reaction was blocked by 3% H_2_O_2_ for 10 min. The ABC kit enhanced the signals of the primary antibody, which was detected by 4-chloro-naphtol. For control staining, PBS replaced either the primary or secondary antibodies and occasionally an irrelevant isotype-specific antibody also employed.

Caspase3 immunocytochemistry was used for identification of apoptotic cells. Infected BF samples were paraffin embedded. After deparaffinization, antigen retrieval with citrate buffer (pH = 0.05%, 10 mM) and permeabilization with PBS containing 1% Tri-ton-X were used. The endogenous peroxidase activity was blocked with 3% H_2_O_2_ (Sigma, H1009) for 10 min The primary antibody was diluted 1:100 in PBS-BSA and the samples were incubated overnight at 4 °C. The secondary antibody, botinylated goat anti-rabbit IgG(H + L) was diluted 1:200 (Vector Laboratories BA-1000) followed by ABC (avidin-biotin-peroxidase complex) (Vectastain Elite ABC kit, Vector Laboratories, PK-6100) for 30 min. After washing, the binding sites of the primary antibody were visualized by DAB (ImmPACT DAB EqV Substrate Kit Peroxidase, Vec-tor Laboratories, SK.4103). After the immunostaining, the samples were counterstained with hemalaun solution.

### 2.5. Glycoprotein Demonstration

Russell modification of Movat pentachrome staining was used [31].

### 2.6. Transmission Electron Microscopy

Tissue samples were fixed in 4% phosphate buffered glutaraldehyde at 4 °C overnight. PBS removed the excess fixative and the samples postfixed in 1% osmium-tetroxide for 2 h. After dehydration in graded ethanol, the bursal tissue was embedded in a mixture of araldite and epoxy-resin (Polysciences, Warrington, PA, USA). Ultrathin sections were contrasted by lead citrate and uranyl-acetate, studied using a Hitachi H-7600 electron microscope.

### 2.7. Image Processing

Images were processed using Adobe Photoshop CC 2017 v.18 (Berkeley, CA, USA).

## 3. Results

### 3.1. Non-Vaccinated, Non-Challenged (NV-NC) Group of Chickens

The BSDCs locate in the medulla of the bursal follicle. Immature and mature (Figure 1a) forms of BSDCs can be recognized. The immature BSDC have few cytoplasmic granules of different sizes, and express FcR [32] that binds maternal IgY. The mature form of BSDCs produces one or two processes and the granules occupy one of the cell processes, but vimentin filaments fill up the whole cytoplasm [32,33]. The location of the granules and the eccentrically locating nucleus endow polarity to the cell. The granules may show an electron lucent spot, at the center of which there is a small electron dense dot. The granules either discharge by exocytosis or fuse together creating large irregular-shaped electron-dense intracytoplasmic corpuscles (Figure 1b). The fused granules may indicate the first sign of BSDC transformation to macrophage-like cells (Mal), which is the result of terminal maturation of BSDCs [17]. The surface of the mature cell, immediately after exocytosis, is covered by a spotted, moderately electron-dense material, which gradually solubilizes in the medulla (Figure 1a). Thus, the granular substance appears in membrane-bound and solubilized forms. The Movat pentachrome staining revealed that the solubilized extracellular substance is a gp (Figure 2a). The amount of gp varies from follicle to follicle, indicating that in the individual follicles the secretory cycle of BSDCs is not synchronized (Figure 2a); therefore, the actual functional stage of the follicles could be different. The mucin-producing interfollicular epithelial cell (IFE) is highly Movat positive, unlike the cells of follicle-associated epithelium (FAE) (Figure 2a). Precursors of BSDCs locate in the CMEA, where they histologically appear as lymphoblasts, large and small lymphocyte-like cells, and they slowly proliferate (Figure 1c). Occasionally, young, immature BSDCs in the mitotic phase are found in the medulla.

### 3.2. Non-Vaccinated, Challenged (NV-C) Group of Chickens

In this group of birds, the Movat staining shows highly variable morphology in the medulla between 36 h and 4 days pi. By day 4 pi, all follicles became smaller (Figure 2b) and between the shrinking follicles a remarkable amount of interfollicular connective tissue (ICT) emerged (Figure 2b). At 36 h pi, the sign of segregation appears as an area containing moderately loosely packed cells in the medulla (Figure 2c) while the cortex seems to be intact. By day 4 pi, the follicular medulla shows variable stages of segregation (Figure 2d,e). First, the segregation shows densely packed cells, to which several Movat-positive Mal may be joined (Figure 2d). The segregated area is isolated from the other part of the medulla, resulting in a “cell-free” area, which is filled with gp (Figure 2e). The surface epithelium became wavy and in the IFE the mucin (gp) production remarkably diminished (Figure 2e), although virus was not present in the cells of IFE. By day 4 pi, the extracellular gp disappeared, and gp emerged in the virus containing Mal. The Movat-positive Mal either join to the segregated area or migrate into the cortex (Figure 2e).

In the early stage of infection (2 days post-infection, pi), the number of infected cells depends upon the virulence of the virus. Much more infected cells were present in the very virulent strain than in the mild and vaccine strains (Figure 3a–c). In the very virulent virus-infected chickens, the central area of the follicular medulla shows a few virus-positive cells (Figure 3a). The majority of 5A mAb-positive cells locate close to the CMEA and at the periphery of cortex. Between these two locations, single, virus-containing cells were found, which may migrate into the periphery of the cortex (Figure 3a). It is necessary to note that there are follicles that show only a few infected cells, if any. The topography of the first infected cells is identical to that of BSDCs (Figure 3b,c). In the mild and vaccine strain infected birds, the virus-containing cells locate mainly in the medulla and only a few cells emerge in the cortex (Figure 3b,c). Many follicles are free of virus-positive cells, but in the mild (Delaware-E) strain, heavily infected follicles may also occur (Figure 3b). In the mild and vaccine strain infected birds, the bursa shows many follicles, which contain only 2–5 scattered virus-positive cells (Figure 3b,c).

The transmission electron microscope shows that the crystalline arrangements of virus particles are packed in the fused BSDC granules, now called corpuscles of Mal. During binding and packing of the virus particles, the BSDCs transforms to Mal and the virus-containing corpuscles appear as large spherical bodies (Figure 3d,e). The rest of the electron-dense substances of corpuscles break up into small nodules and myelin figures, which possibly come from BSDC granules. The actual size of a virus-containing corpuscle can be 5–6 microns, which size is identical to that of a small lymphocyte and at the periphery of Mal a remarkable number of neutral lipid droplets accumulate (Figure 3d). In the corpuscles, the virus particles gradually become hazy and indistinct (Figure 3e). In the segregated area of several follicular medulla, the caspase reaction reveals accumulating apoptotic cells (Figure 3f,g). In other follicles, the segregated area shows many newly formed BSDCs, besides the virus containing Mal, demonstrating that individual follicles are in different functional stages (Figure 3f).

### 3.3. Vaccinated Non-Challenged (V-NC) Group of Chickens

The Movat staining shows a remarkable difference between the early vaccine-take (Figure 4a–c) and chronic stage of bursa (Figure 4d,e). In early-stage vaccine-take group of chickens, the most remarkable feature is the absence of extracellular gp, the numerous moderately Movat-positive Mal, and the decreased number of BSDC precursors in the CMEA, the cortex is highly depleted and there is no neutrophil granulocyte and monocyte invasion (Figure 4c). The CMEA mainly transformed to a cuboidal-shaped epithelial cell layer because of the majority of BSDC precursor cells are gone, they differentiated to immature BSDCs. The bursal folds are moderately shrunken, the follicles are close to one another and the amount of ICT somewhat increased (Figure 4a). The surface epithelium of the fold is slightly wavy, and the mucin (gp) production decreased in the IFE. Mucin appears only in the most apical portion of IFE cells (Figure 4a,b).

In the chronic stage of bursa, there are many transitory forms, which depend on the number of the rest of the BSDC precursors in the CMEA and the amount of extracellular gp (Figure 4d,e). In lesser affected follicles, the extracellular gp remains, indicating that the structural integrity of stromal components and a considerable number of BSDC precursors remained intact, allowing B-cell recovery. The CMEA shows variable structure: that is, in some places the BSDC precursors are preserved (Figure 4e and Figure 5a), while in other places the CMEA is “empty”, namely, there are no BSDC precursor cells (Figure 4e and Figure 5b), the CMEA collapsed.

### 3.4. Vaccinated, Challenged (V-C) Group of Chickens

These birds were inoculated with the vaccine 228E attenuated virus strain at 28 days of age, when the vaccine virus started to replicate in the bursa, initiating the onset of the active immune response. The general histological structure of the BF basically does not differ from that of the V-NC group of birds (compare Figure 4a,f). The extracellular gp is absent (Figure 4g,h), the number of Mals/follicles is variable, from a few Mal(s) (Figure 4g) up to a group (Figure 4h). Generally, the number of Mal/follicles is remarkably less than in the V-NC group (compare: Figure 4c,g). The CMEA collapsed and the arch forming cells showed a cuboidal-shaped epithelial layer like in V-NC group and the shape of the epithelial cell nuclei is highly bizarre (Figure 5b).

## 4. Discussion

In the medulla of the bursal follicle, only the BSDCs have classical secretory machinery around the cytocentrum. There is a well-developed Golgi complex with secretory granules and a considerable number of free ribosomes, but the granular endoplasmic reticulum (GER) is moderately developed. It is well-known that the structural, non-secreting proteins are made on free ribosomes—like the myofibrils in myoblasts—and the GER produces a secretory substance in plasma cells or exocrine pancreatic cells. In BSDCs, the proportion of free ribosomes and GER suggests that the BSDCs may be responsible for two kinds of product and function. The function of BSDCs may be found within the fate of secretory granules, which either discharge or fuse together forming large, irregular-shaped, dense bodies, corpuscles: (a.) In the case of the granular discharge, the secreted substance appears as a spotted shape, attached to the BSDC membrane, where the granular discharge is in a high concentration, which possibly makes the BSDCs a primary target of IBDV. This statement may be confirmed, that in the medulla, the scattered appearance of the first IBDV-positive cells is identical with the topography of the BSDCs. From this membrane-bound form, the granular substance gradually detaches, solubilized in the extracellular space of medulla contributing to the ME of follicular medulla [17,18]. The solubilized, extracellular gp binds to the medullary B lymphocyte [19] in a much lower concentration; therefore, the IgM surface positive B cells are the secondary target cells of IBDV. Possibly, the membrane-bound gp on medullary lymphocytes is necessary for maintenance of B cell survival and may participate in the binding of the serotype I IBDV strain [20]. (b.) In normal uninfected birds, the BSDC granules fuse together, the BSDCs transform to Mal, which is the result of terminal maturation of BSDCs. The Mal either enters the FAE [34,35] or occasionally migrates to the cortex. In the case of IBDV infection (NV-C), the Mal, carrying the virus, enters the cortex, inducing inflammation—in infected birds the migratory pathway of Mal is shifted from the FAE to cortex. In the vaccinated (chronic stage) challenged (V-C) group of birds, mainly heterophil invasion occurs. A significant number of heterophil granulocytes enter the IFE. Depletion of heterophil granulocytes, by 5-fluorouracil, prior to IBDV infection resulted in only mild clinical symptoms. These observations show that in addition to the BSDCs, the neutrophil granulocyte is the major responsive cell for the IBDV infection [36]. The virus uptake and packaging in the corpuscle of Mal inhibits the granular discharge resulting in the absence of gp in the extracellular space of medulla, and subsequently B cell apoptosis. The Mal either accumulate in the segregated area of the medulla or migrate into the cortex inducing an inflammatory reaction, and they are eliminated. Other segregated areas show many newly formed, virus-free BSDC.

The non-vaccinated and non-challenged (NV-NC) control group of birds reveals extracellular gp in the follicular medulla. The amount of extracellular gp is highly variable in the follicles [17,18], indicating some functional stage differences between the follicles. The reason for these functional differences is not known, but the actual number of mature BSDCs can contribute to the gp differences in the follicles. In the early stage of vaccine virus replication (V-NC), the extracellular gp disappears or decreases below the sensitivity of Movat staining, but in the chronic-regenerating phase, gp reappears in the follicles indicating the structural integrity of stromal elements [37] and the rest of the BSDC precursors, allowing the functional or at least partially histological restoration of the bursa. In the (V-NC) group of chickens, the migration pattern of Mal is more similar to that of uninfected control birds, namely, the majority of Mal enters the FAE [38] and is eliminated in the bursal lumen. The limited number of Mal, which migrate into the cortex, is not enough to induce inflammation, because there is no monocyte, heterophil invasion.

The cell surface of chicken embryo fibroblasts (CEFs) is also covered by gp [17], which makes the CEF a common tool for IBDV propagation. If in the case of CEF the membrane-bound gp has a role in the virus binding and uptake, then the presence of extracellular gp should have a crucial role in IBDV infection. The vaccination remarkably diminished or closely abolished the extracellular gp in the medulla. In birds, infected with field IBDV, the bursal lesion is highly variable between 36 h and 4 days pi, which is also valid for the clinical signs. Rauw et al. [39] reported that the overproduction of ChINFϒ by T cells has a key role in long-lasting immunosuppression. In place of bursal lesions, T cells must be accumulated in a sufficient number for ChINFϒ production. The ChINFϒ activates macrophages to produce inflammatory factors, which further results in overproduction of ChINFϒ in the bursa but not in the blood [39]. In normal bursal follicle, T cells rarely occur; therefore, from the bursal lesions some type of factor(s) should be released to activate T cells to migrate into bursal lesions and produce ChINFϒ, which activates macrophages. The cellular interaction between T cells and macrophages works in the adaptive immune response, but the innate immune response is prompt and does not require time-consuming cell-to-cell collaboration. The Mal as a virus containing possibly activated macrophage, is capable of inflammatory cytokine production, such as ChIL-6 (and other cytokines), initiating an innate immune response. In the (NV-C) group of chickens, the Mal, which is a transformed BSDC carrying the virus, migrate to the cortex, and induce inflammation by recruiting T cells, monocytes, and heterophil granulocytes. The virus-containing cell can enter the circulation and settle down in the thymus [40], suggesting that the circulation can contribute to the virus dissemination.

It is thought that the virus is released after cell lysis. The VP5 (viral protein) accumulates in the plasma membrane of infected cells and modifies the morphology and membrane permeability of host cells [41]. The altered K^+^ current may delay the apoptotic process and cell lysis [42]. Therefore, the cell lysis theory for viral dissemination may be challenged. The virus is wrapped in the former BSDC granules (now corpuscles of Mal), which represents the first, non-lytic, early phase of virus dissemination [43]. The virus containing Mal may migrate to the FAE, but mainly into the cortex, where the cells can enter the circulation [40]. The late or second phase of virus replication could be related to lymphocyte infection and apoptosis; cell lysis. The lymphocyte apoptosis can be induced by lack of extracellular gp [17,18] and the second phase of B cell infection [43].

At hatching, the number of BSDCs/follicles is 10 ± 4, and by day 14 it is increased to 53 ± 11, while at 70 days of age birds reach 197 ± 23 [33]. Rautenschlein and Haase [11] did not find age-related differences in outbreak of clinical disease between 12 and 28 days pi. By 14 days post-hatching, the number of BSDCs increased more than five times by day 14, which may be enough to initiate clinical disease. However, Rautenschlein and Haase [11] reported that the chickens recovered from in ovo vaccination faster than the post-hatched vaccinated birds. Around hatching, the BSDCs develop FcR [32], that bind maternal IgG. Thus, in the in ovo vaccinated birds, the BSDCs are still underdeveloped; therefore, the recovery may be faster than post-hatched vaccinated birds, when the FcR of BSDCs already binds maternal IgG.

The intracellular, Movat-positive gp appears in every infected group of chickens, while in the NV-NC group the gp is in the extracellular space of the medulla. The VP5 is a non-structural protein of the IBDV [44], and incorporated in the membrane of infected cells, changing the K^+^ ion current [42]. The VP5 protein alters the cell morphology and membrane permeability [41]. These findings raise the issue that in the Mal, the gp may originate from the pinocytosis of the extracellular gp. However, in the vaccine-take group of chickens, the amount of extracellular gp remarkably decreases below sensitivity of Movat staining, and some remaining extracellular gp becomes s solid, electron-dense mass, close to the FAE [18]. Thus, in the V-C group of birds, the gp in the Mal cannot come from the uptake of extracellular gp. However, these observations raise two possibilities, because the gp cannot be seen in the BSDCs of control, normal birds. One option is that the BSDCs produce two chemically and functionally different granules: (a.) The granules, which discharge contain “hidden” extracellular gp. (b.) The other granules, which fuse together consist of classical macrophage lysosomes. The other possibility is that during solubilization of granular contents undergo certain changes in molecular configuration and the extracellular gp becomes available for Movat staining. If this is true, then the effect of virus wrapping can also change the molecular configuration, which results in the appearance of Movat-positive gp in the Mal. Thus, the determination of granular contents is crucial for further studies.

## 5. Conclusions

BSDCs produce secretory granules, which either fuse together or discharge. During physiological maturation, the BSDCs transform to Mal and enter the FAE. The granular discharge results in extracellular gp, which contributes to the medullary microenvironment and binds to the medullary lymphocyte. The gp is the first recognized component of the medullary microenvironment. IBDV is taken up by BSDCs and wrapped up into the secretory granules and replicates. The virus packaging into the BSDC granules—now called corpuscles of Mal—prevents the granular discharge resulting in the absence of extracellular gp, and subsequently alteration in the medullary microenvironment. Movat-positive gp emerges in the virus containing Mal. Electron microscopic analysis showed that the turnover of the glycoprotein is associated with the life cycle of BSDCs. The Movat-positive extracellular gp could be a good indicator of the immunological status of the chicken. The follicle can have more mature than immature BSDCs and abound in extracellular fine-flocculated substance, namely gp and vice versa. If the number of immature BSDCs is high and the granular discharge is moderate, the Movat-positive gp is low; therefore, the medulla shows only weak or no fine flocculated substance, i.e., gp. Post-vaccination, the solubilization of BSDC discharge is restricted and accumulates in huge masses on the surface of BSDCs (paper is under preparation). It is surprising, that these extracellular masses are not Movat positive, and possibly not “available” for IgM-positive, medullary B cells. The identification of the exact molecular nature of the gp and the biological function of the glycoprotein in B cell proliferation and/or survival requires further work.

## Figures and Tables

**Figure 1 viruses-14-01689-f001:**
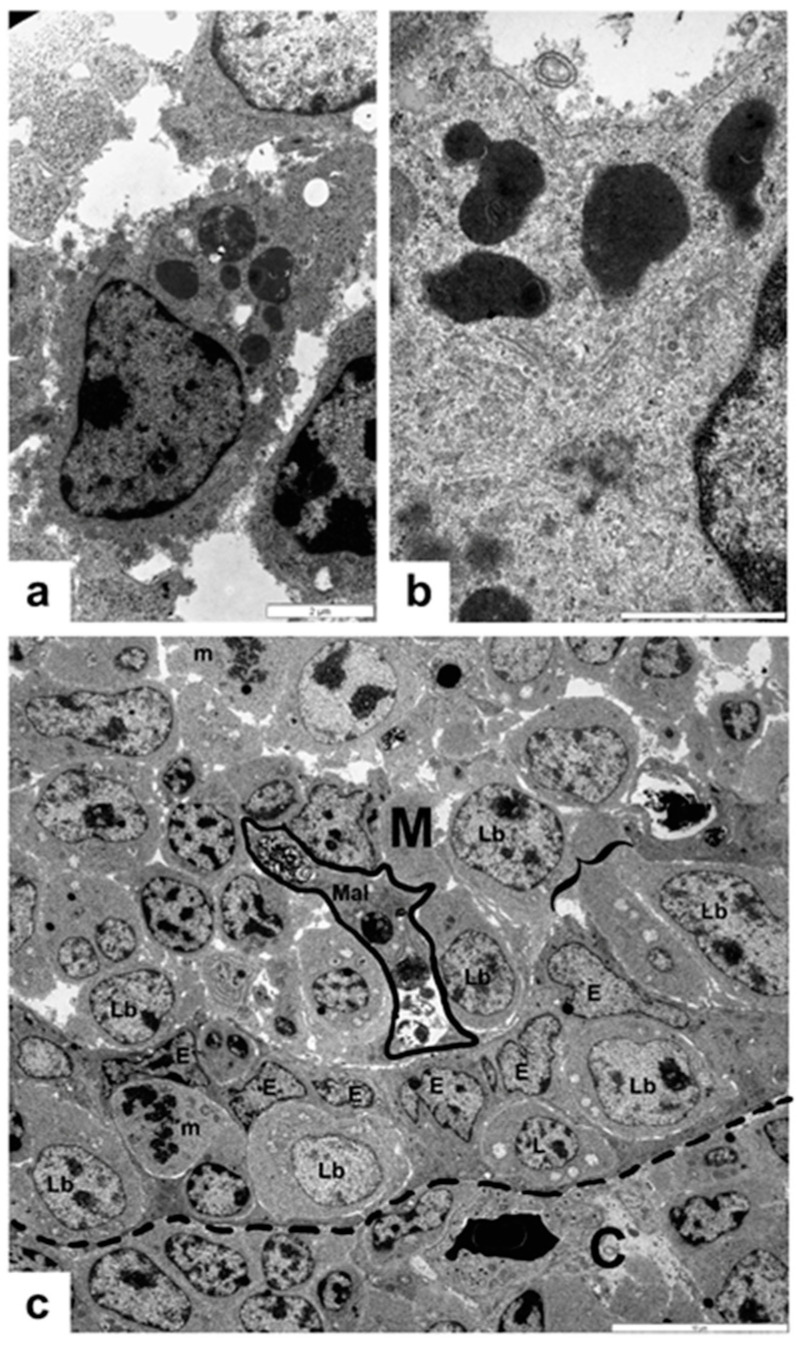
Transmission electron microscopy showing the BSDC and cortico-medullary (CM) border. (**a**) Mature BSDCs. The eccentrically located nucleus and the electron-dense granules collect in one of the cell processes endowed with polarity of the BSDCs. The substance of discharged granules attaches to the BSDC membrane. (**b**) The irregular-shaped, fused cytoplasmic granules are around the Golgi zone and centriole (C). (**c**) In the cortico-medullary epithelial arches (CMEA), lymphoblasts (Lb), medium-sized lymphocyte-like cells (L), and mitotic figures (m) are found. The clamp shows the opening of a CMEA and two lymphoid-like cells enter the medulla Basal lamina: dashed line; E: cell of CMEA; M: medullary; C: cortex; m: mitosis; Mal: macrophage-like cell (with apoptotic lymphocytes outlined).

**Figure 2 viruses-14-01689-f002:**
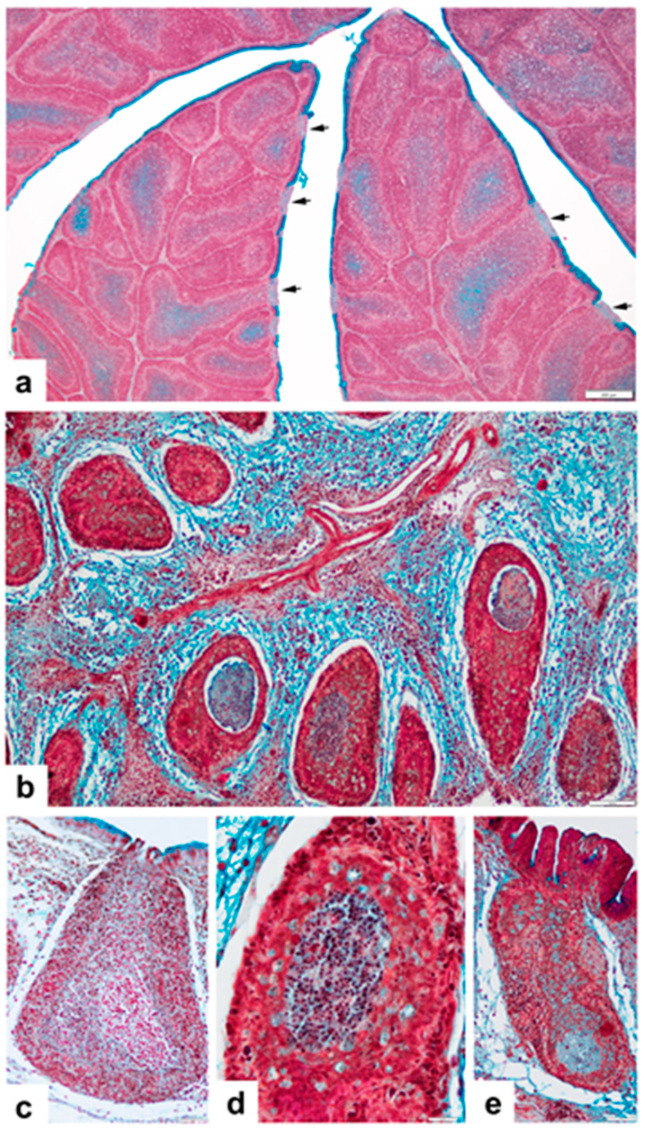
Movat pemtachrome staining of bursa of Fabricius (**a**) Non-vaccinated, non-challenged (NV-NC) group of chickens. The cells of IFE and FAE are Movat positive and negative (arrows), respectively. In the medulla, the staining intensity of extracellular gp is variable. (**b**) Non-vaccinated, challenged (NV-C) group of chickens. Four days pi, a very virulent IBDV-infected bird. Detail of a bursal fold. The follicles shrank, in several follicular medulla, different stages of segregation emerged, and the interfollicular connective tissue (ICT) remarkably increased. (**c**) At 36 h pi. The follicular medulla shows an area (outlined) where the cell density may be lower. The cortex is intact. (**d**) Four days pi. The segregated area is densely packed with cells and large, Movat-positive cells (Mal) found around the segregated area. Cortex is highly depleted. (**e**) The surface epithelium is wavy, and the gp production ceased in the IFE, but a large number of Movat-positive Mal(s) occupy the medulla. Mal may be fused with the segregated area which is filled with a Movat-positive substance. Mal are also present in the depleted cortex.

**Figure 3 viruses-14-01689-f003:**
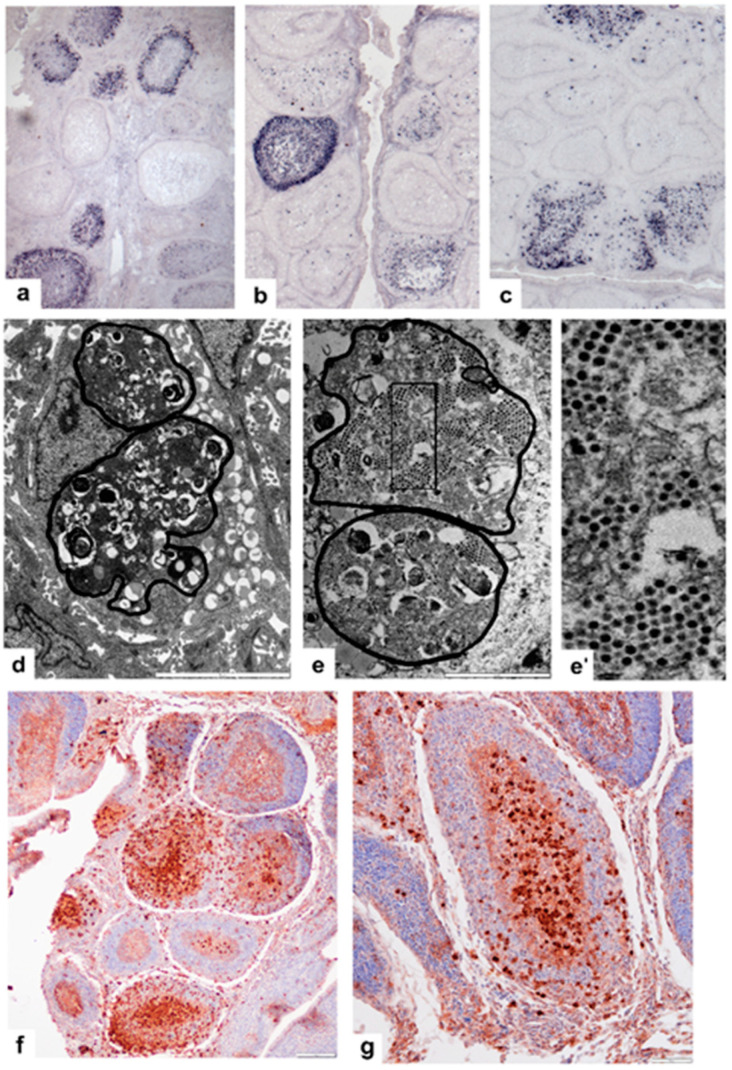
Immunocytochemistry of anti-IBDV; 5A10 mAb, 2 days pi. (**a**) In the very virulent virus infected birds, the majority of 5A10 positive cells are located close to the CMEA(s) and they are able to assemble in the periphery of the cortex. Many individual cells are between the CMEA(s) and periphery of the cortex. Follicles are still found without infected cells. (**b**) The Delaware-E strain shows much fewer infected follicles, but occasionally a heavily infected follicle occurs, which differs from that of very virulent virus infected follicles. Many IBDV-positive cells are in the medulla and the cortex is homogenously covered by IBDV-positive cells. (**c**) The vaccine 228E virus positive cells, seem to be similar to that of the Delaware-E strain. (**d**) Macrophage-like cells (Mal) from a vaccine-strain virus inoculated chicken. The cell has two corpuscles (fused BSDC granules, outlined) and the periphery of the cytoplasm is full of neutral lipid droplets. (**e**) Higher magnification from a Mal. The corpuscles (outlined) are full of virus particles and the dark nodules and myelin—the rest is corpuscles material. (**e′**) is an inset of (**e**). The rectangle shows higher magnification of the virus particles. (**f**) At 36 h pi, the caspase3 staining shows a highly variable number of apoptotic cells in the follicles. (**g**) Higher magnification shows that the caspase3-positive cells are found in the medulla and a few cells occur in the periphery of the cortex.

**Figure 4 viruses-14-01689-f004:**
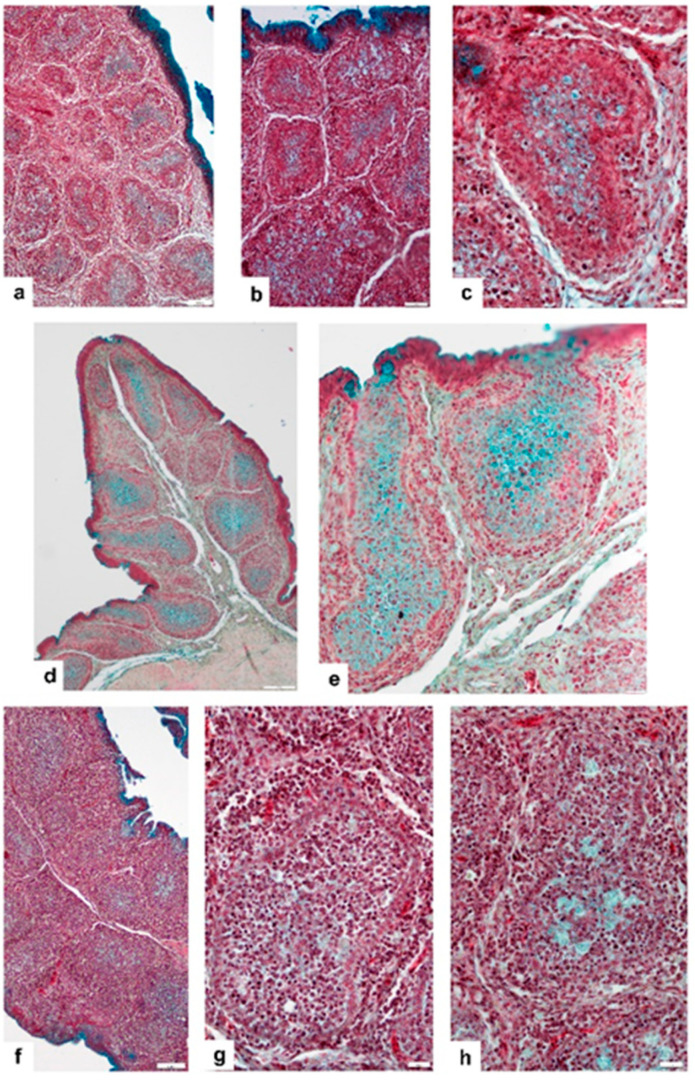
Vaccinated, non-challenged (V-NC) group of chickens sampled at D31 (3 dpv) (**a**–**c**) and at D35 (7 dpv) (**d**,**e**), representing the acute phase and the chronic stage of onset of vaccine virus replication, respectively. (**a**) The follicles are closely packed in the fold and the medulla contains many Movat stained Mal. (**b**) The surface epithelium of the fold is moderately wavy, mucin (gp) is found only in most apical part of IFE cells. Some follicles maybe had a small amount of extracellular gp. (**c**) The major part of CMEA consists of cuboidal-shaped epithelial cells. The cortex is thin and depleted. ICT moderately increased. (**d**) The most remarkable finding is the presence of extracellular gp in the follicles. (**e**) Higher magnification shows that in some places, the CMAE is well-retained, contains BSDC precursors (arrows). (**f**–**h**) Vaccinated, challenged (V-C) group of chickens. (**f**) The fold is densely packed with shrunken follicles, otherwise the histological structure is similar to that of the chronic stage group of birds. (**g**) The cells of CMAE transformed to cuboidal-shaped epithelial cells (arrow). In the medulla, several moderately stained, Movat-positive Mal(s) may be found. (**h**) The follicles have few aggregated Movat-positive Mal(s), but generally their number is lower than in the vaccine-take (V-NC) birds.

**Figure 5 viruses-14-01689-f005:**
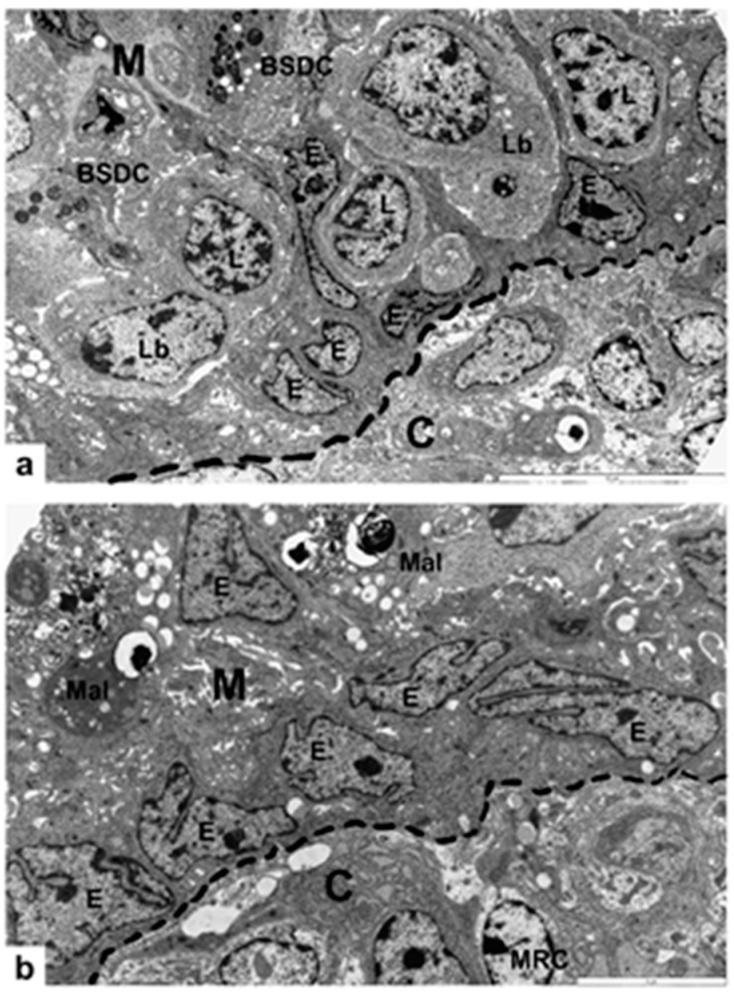
Transmission electron micrographs taken from the V-NC group of chickens. (**a**) This portion of CMEA retained the BSDC precursors. C: cortex; M: medulla; E: epithelial cell of CMEA; Lb, L: lymphoblast and lymphoid-like cells; dashed line: cortico-medullary bursal lamina. (**b**) A collapsed portion of CMAE, where BSDC precursors are absent and the arch-forming epithelial cells transformed to cuboidal-shaped cells. M: medulla; C: cortex; E: epithelial cells of collapsed CMEA; Mal: macrophage-like cell; MRC: mesenchymal reticular cell; dashed line: cortico-medullary basal lamina.

**Table 1 viruses-14-01689-t001:** Outline of the animal phase of the trial. Dates indicate the age of the chickens at the given action in days. Abbreviations: days post-vaccination (dpv) and days post challenge (dpch).

	Group	Number of Chickens	Vaccination	Challenge Infection	Sampling
1.	NV-NC (control)	5	No	No	At D35
2.	NV-C-a (vvIBDV)	15	No	at D31	At D32.5 (36 h post-infection, at D33 (48 h p-i) and at D35 (4 dpch)
	NV-C-b (Delavare E)	15	No	at D31	At D32.5 (36 h post-infection, at D33 (48 h p-i) and at D35 (4 dpch)
3.	V-NC	10	at D28	No	At D31(3 dpv) and D35 (7 dpv)
4.	V-C-a (vvIBDV)	5	at D28	at D31	At D35 (4 dpch)
	V-C-b (Delavare E)	5	at D28	at D31	At D35 (4 dpch)

## Data Availability

Not applicable.

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
