# Peer review of "Glycoprotein Production by Bursal Secretory Dendritic Cells in Normal, Vaccinated, and Infectious Bursal Disease Virus (IBDV)-Infected Chickens"

_viruses, 2022, doi:10.3390/v14081689_

Round 1

Reviewer 1 Report

This paper compared and studied the effect of IBDV infection on the gp producing BSDC and the alteration of Movat positive, bursa specific gp in mock, infected, vaccinated, and vaccinated plus challenged chickens. Results showed that BSDC produces secretory granules, which either fuse together or discharge. During physiological maturation, the BSDC transforms to Mal and enter the FAE. The granular discharge results in extracellular gp, which contributes to the medullary microenvironment and binds to the medullary lymphocyte. IBDV is taken up by BSDC and wrapped up into the secretory granules and replicates. The virus packaging into the BSDC granules (corpuscles of Mal) and prevents the granular discharge resulting in absence of extracellular gp, and subsequently alteration in medullary microenvironment. These data indicated that the BSDC contributed more to IBDV infection. This study is very important for further understanding the histological and cytological mechanism of IBDV infection. Here are some suggestions for revision.

1. In abstract, some necessary factors of this study are missing, including methods, results.

2. Is the data presented in this article statistically significant?

3. Is it necessary to summarize the main differences among the experimental groups in the form of tables.

4. Line 419, the turnover of the glycoprotein is associated with the life cycle of BSDC. Can you expand the discussion a little?

5. Line 419-420, the Movat positive extracellular gp could be a god indicator of the immunological status of the chicken. Can you explain it in detail?

Author Response

Reviewer 1

1.)          Figs 3. We added an inset of virus particles with higher magnification. Fig. 4. It is difficult to improve, because the paraffin sections are 5-6 um and the Movat staining has five different dyes, which “torture” the section.

2.)          The typing errors are improved.

3.)          The Materials and Methods is subtitled.

4.)          The virus strains are rewritten. The lines 101-110 is cancelled and by a new description of the virus strains replaced. Therefore three new references must have been added (28; 29; 30).

5.)          Hours and minutes changed as the reviewer proposed.

6.)          Is the research design appropriate? We have published the gp in normal and infected birds (Felföldi et al. 2021) as well as the transformation of BSDC to Mal (Oláh et al. 2022). The recent paper focuses for the changes in the extracellular gp in vaccinated and vaccinated plus challenged birds. Therefore, the research design may be looked-like to be inappropriate. We have to apologize for it. But only one of the reviewer said, that the research design is not appropriate. This may be some excuse for our method. 

7.)          Are the methods adequately described? Almost every description could be better or improved, but we followed the histological changes after vaccination and vaccination plus challenge by anti-VP2 and anti-Caspase3 immunohistochemistry and transmission electron microscopy. The gp changes in cells of IFE and in the extracellular space of medulla were followed by Movat pentachrome staining. Therefore, we believe that the applied methods are satisfactory for the results and supports the conclusion.

Reviewer 2 Report

Infectious bursal disease virus (IBDV) is a well-known immunosuppressive pathogen in chicken which has caused huge economic losses to the poultry industry worldwide. However, the pathogenesis of IBDV is unclear. In this study, Felföldi et al. showed the effect of IBDV infection on the glycoprotein (gp) producing bursal secretory dendritic cells (BSDC) and the alteration of Movat positive, bursa specific gp in infected, vaccinated and vaccinated plus challenged chickens. The data in this study is interesting and can support their conclusion. However, there are still some points should be improved before accepted for publication.

1. Figures 3 and 4 shown here are less clear than the other figures in the Ms. Please improve the quality of them.

2. There are many unnecessary “-” within or between the words. Such as: Line29, clinical; Line31, replicate; Line38, experiments; Line47, prevent; Line305, “in myoblasts –” should be “in myoblasts”; and so on. Please correct them in the Ms.

3. In the 2. Materials and Methods, subtitle should be added in this part. Such as: 2.1 Animal experiments; 2.2 Immunocytochemistry; and so on.

4. In Line 106-107: First, please add an abbreviation for very virulent IBDV (vvIBDV). In addition, it would be better to give a brief background of the IBDV strains used in this study. The authors could directly cite a reference here. Last, Delaware-E strains is not consistent with that in Table 1.

5. It would be better to use “min”, “h” instead of “minutes”, “hours” in this Ms.

6. The authors used dpv and dpch to represent days post-vaccination and days post challenge. So, why not just use dpi and hpi to represent day post-infection and hour post-infection in this Ms?

Minor errors as follow:

1. Line 69: „immature” should be “immature”.

2. Line 115-116: please use “The tissue samples from each chicken” instead of “From each chickens the tissue samples”.

3. Line 130: please use “℃” instead of “oC.”.

4. Line 144: Please delete the additional full stop at the end of the sentence.

5. Line 147-166: the written type in this part is different from the others.
6. Line 200: It would be better to correct the sentence here into “The surface epithelium became wavy, and the mucin (gp) production remarkably diminished in the IFE”.

7. Line 303: please use “considerable amounts of” instead of “considerable amount of.”

8. Line 360: “somekind” should be “some kind”

Author Response

Reviewer 2

1.)          In abstract, some methods and results are missing. In the Abstract, the Movat staining is mentioned, but the immunocytochemistry and transmission electron microscopy are missing. We added two sentences to the Abstract. The majority of the Abstract is the result.

2.)          The immunohistochemical results are mainly qualitative and the low number of samples does not allow statistical analysis.

3.)          To summarize the differences among the group in form of table would be useful, however in this paper; it would be a little bit redundant.

4.)          Is the research design appropriate? We have published the gp in normal and infected birds (Felföldi et al. 2021) as well as the transformation of BSDC to Mal (Oláh et al. 2022). The recent paper focuses for the changes in the extracellular gp in vaccinated and vaccinated plus challenged birds. Therefore, the research design may be looked-like to be inappropriate. We have to apologize for it. But only one of the reviewer said, that the research design is not appropriate. This may be some excuse for our method. 

5.)          Are the methods adequately described? Almost every description could be better or improved, but we followed the histological changes after vaccination and vaccination plus challenge by anti-VP2 and anti-Caspase3 immunohistochemistry and transmission electron microscopy. The gp changes in cells of IFE and in the extracellular space of medulla were followed by Movat pentachrome staining. Therefore, we believe that the applied methods are satisfactory for the results and supports the conclusion.
